# The impact of local assembly rules on RNA packaging in a *T* = 1 satellite plant virus

**Sam R. Hill**[1,3], **Reidun Twarock**[1,2,3], **Eric C. Dykeman**[1,3]*

**1** Department of Mathematics, University of York, York, United Kingdom, **2** Department of Biology, University of York, York, United Kingdom, **3** York Cross-Disciplinary Centre for Systems Analysis, University of York, York, United Kingdom

* eric.dykeman@york.ac.uk

**Data Availability Statement:** Software for running RNA packaging simulations is available via github at: https://github.com/edykeman.

**Funding:** RT acknowledges funding via an EPSRC Established Career Fellowship (EP/R023204/1), a

## Abstract

The vast majority of viruses consist of a nucleic acid surrounded by a protective icosahedral protein shell called the capsid. During viral infection of a host cell, the timing and efficiency of the assembly process is important for ensuring the production of infectious new progeny virus particles. In the class of single-stranded RNA (ssRNA) viruses, the assembly of the capsid takes place in tandem with packaging of the ssRNA genome in a highly cooperative co-assembly process. In simple ssRNA viruses such as the bacteriophage MS2 and small RNA plant viruses such as STNV, this cooperative process results from multiple interactions between the protein shell and sites in the RNA genome which have been termed packaging signals. Using a stochastic assembly algorithm which includes cooperative interactions between the protein shell and packaging signals in the RNA genome, we demonstrate that highly efficient assembly of STNV capsids arises from a set of simple local rules. Altering the local assembly rules results in different nucleation scenarios with varying assembly efficiencies, which in some cases depend strongly on interactions with RNA packaging signals. Our results provide a potential simple explanation based on local assembly rules for the ability of some ssRNA viruses to spontaneously assemble around charged polymers and other non-viral RNAs *in vitro*.

## Author Summary

Assembly in single-stranded RNA plant viruses takes place via a highly cooperative process in which capsid proteins co-assemble around ssRNA. In the small satellite plant virus STNV, small hairpins present in the genome, termed packaging signals, bind to capsid proteins during assembly and allow for efficient formation of the capsid shell. Although these hairpins have been visualised in X-ray crystallography, the local rules of their interaction with the capsid proteins and how they ensure an efficient assembly process is somewhat unknown. Here we test several assembly scenarios involving different local rules for the protein-protein and RNA-protein interactions and find that assembly efficiency is highly dependent on the local assembly rules. Interestingly, while certain local assembly rules are consistent with a packaging signal mediated assembly model, some local rules

Royal Society Wolfson Fellowship (RSWF/R1/ 180009), and a Joint Wellcome Trust Investigator Award with Peter Stockley from the University of Leeds (110145 & 110146) The funders had no role in study design, data collection and analysis, decision to publish, or preparation of the manuscript.

**Competing interests:** The authors have declared that no competing interests exist.

predict reasonable assembly efficiency independent of packaging signal distribution. This may explain the ability to package charged polymer materials in some plant viruses.

## Introduction

Assembly in the class of plus sense single-stranded RNA (+ssRNA) viruses occurs via a complex cooperative assembly process where the ssRNA genome is spontaneously surrounded by the protective protein shell (i.e. the capsid). Unlike the assembly process in double-stranded DNA (dsDNA) viruses which utilise a viral terminase protein to pump dsDNA into a pre-formed capsid shell using ATP as an energy source, assembly in +ssRNA viruses is spontaneous, with the energy required to fuel the process coming from both protein-protein interactions as well as interactions between the genome and protein shell. *In vivo*, the packaging process in +ssRNA viruses is highly specific, with viral capsids preferentially assembling around viral RNA, with much lower efficiency for assembly around cellular messenger RNAs [1]. The assembly process is further complicated in +ssRNA viruses by the fact that the RNA genomes in these viruses must play multiple roles during the infection process [2]. Specifically, the +ssRNA genome must act; (1) as a template for production of viral proteins by the host ribosome, (2) a template for minus strand synthesis by the viral RNA dependent RNA polymerase (RdRp), and (3) a substrate to be packaged into new viral particles. Regulation and timing of these multiple roles is expected to be critical to ensuring high yield of viral particles as pre-mature assembly of ssRNA will lower viral protein production by removing RNAs being translated by host ribosomes.

Recent high-resolution cryo-EM experiments on ssRNA viruses [3–5] have allowed for the packaged RNA to be visualised, providing further clues as to how the assembly process occurs in this class of viruses and how it may be regulated during infection of a host cell. In Koning et al. [3] an asymmetric cryo-EM reconstruction of bacteriophage MS2 at approximately 3 Angstrom resolution has revealed multiple RNA hairpin structures in contact with the capsid shell. These hairpins have common structural and sequence similarities with the translational repressor (TR) hairpin which, in addition to promoting a conformational switch in the capsid protein (CP) that promotes assembly [6], allows for coat proteins to act in a gene regulatory role by repressing synthesis of RdRp when CP is bound to TR [7]. These additional hairpins (aside from TR) promote the conformational switching of coat proteins during assembly, guiding formation of the capsid shell and formation of the six five-fold symmetry axes of the capsid. Moreover, a stochastic assembly model demonstrated that the presence of a high affinity hairpin site such as TR within the viral genome was sufficient to ensure nucleation of assembly on viral mRNA over competitor host mRNAs [1]. In contrast to the fairly ordered cryo-EM study of MS2, a recent cryo-EM structure of *in vivo* assembled Brome mosaic virus particles (BMV), a plant virus which is related to the cowpea chlorotic mottle virus (CCMV), has surprisingly revealed an RNA density which seems largely disordered, with preferential RNA-protein contacts at three-fold and two-fold axis of the capsid [4]. However, since BMV and CCMV capsids are capable of packaging charged polymer materials, such as polystyrene sulphate, under certain conditions *in vitro* [8, 9], the disordered RNA density is potentially due to these viruses adopting a non-specific assembly mechanism of charge neutralisation.

These observations in simple plant (BMV/CCMV) and bacterial (MS2/Q$\beta$) systems have led to several theoretical models of the assembly process in ssRNA viruses [1, 10–14] and the distribution of RNA inside the capsid shell. From this theoretical and experimental work, two main paradigms exist to describe possible ways in which the ssRNA genome can cooperatively

co-assemble with the capsid shell: a non-specific model, and a packaging signal-mediated model. In the non-specific model which has been used by several groups to describe assembly in the plant viruses BMV and CCMV [9], interactions between the negatively charged phosphate backbone of RNA and the positively charged arginine/lysine rich N-terminal arms present in these viruses, contributes to virus assembly and partial neutralisation of the charge. This mode of assembly has been examined theoretically by Brownian dynamics models of assembly [12] which include an additional term in the potential energy model accounting for protein-RNA interactions. Since these interactions are non-specific, capsid assembly in these viruses would presumably need to take place in viral factories, i.e. compartments in the cell where high concentrations of viral RNAs are present over host mRNAs, in order to prevent assembly around host mRNAs. In contrast, the packaging signal-mediated model proposes that specific RNA secondary structures with high-affinity for CPs trigger the assembly of the capsid shell, and their specificity for CPs insures selective packaging of viral RNAs over host mRNAs [1]. In addition to promotion of selective packaging, packaging signals (PSs) can play additional roles in the regulation of the viral life cycle, as exemplified by the TR hairpin in MS2 and Q$\beta$, which suppress RdRp synthesis in response to high CP concentrations. Other non-regulatory PSs can have heterogeneous affinities for CPs, which may aid in avoiding kinetic trapping of the RNA during assembly [10].

Unlike the plant viruses BMV and CCMV which have shown mostly disordered RNA in cryo-EM images, high resolution X-ray imaging of the plant virus STNV has shown well ordered RNA helix structures regularly placed near the two-fold axes of the capsid [15]. Interestingly, the STNV CP has a positively charged, N-terminal domain, similar to the plant viruses CCMV and BMV, which may suggest non-specific interactions are driving assembly in this virus. However, several experimental studies have shown that PS:CP interactions result in charge neutralisation of the N-terminal alpha helices, allowing formation of the three-fold axes of the capsid [16]. Moreover, recent experiments have also demonstrated that altering specific sequence and structural motifs in the 5' UTR of STNV genome affects packaging efficiency, implying that STNV utilises a packaging signal-mediated assembly mechanism [2]. In this work, we demonstrate using stochastic modelling that a packaging signal-mediated assembly model for STNV assembly, combined with a specific set of local set of assembly rules on the RNA connections between CPs, results in efficient assembly. We suggest a range of choices of local rules, generated from the protein structure of the capsid, that defines which protein positions can be bound to sequential PSs on the RNA. The local rules were chosen using rotations about the four nearest symmetry axes for the proteins, insuring that symmetry is maintained, e.g. if the RNA can go clockwise around its closest 5-fold axis, then it can also go anticlockwise. Interestingly, only certain local rules are compatible with highly efficient assembly (>90% of capsids assembled) resulting in a highly conserved RNA organisation that is strongly dependant on PS distribution. Other local assembly rules, however, can result in assembly with very disorganised RNA structures and weaker dependency on PS distribution. We discuss the implications of the local assembly rules in describing assembly in ssRNA plant viruses.

## Methods

### Structure of the STNV capsid

The STNV capsid is a small *T*=1 icosahedral shell composed of 60 copies of a 21 kDa coat protein [17] (c.f. Fig 1a). Icosahedrally averaged X-ray crystallography of STNV capsids, crystallised in the presence of a small RNA hairpin (Fig 1b) previously identified via SELEX [18], has revealed density of RNA near the two-fold axes of the capsid in the form of a 3-bp A-form RNA helix [15]. The positioning and icosahedral averaging of the capsid during structure

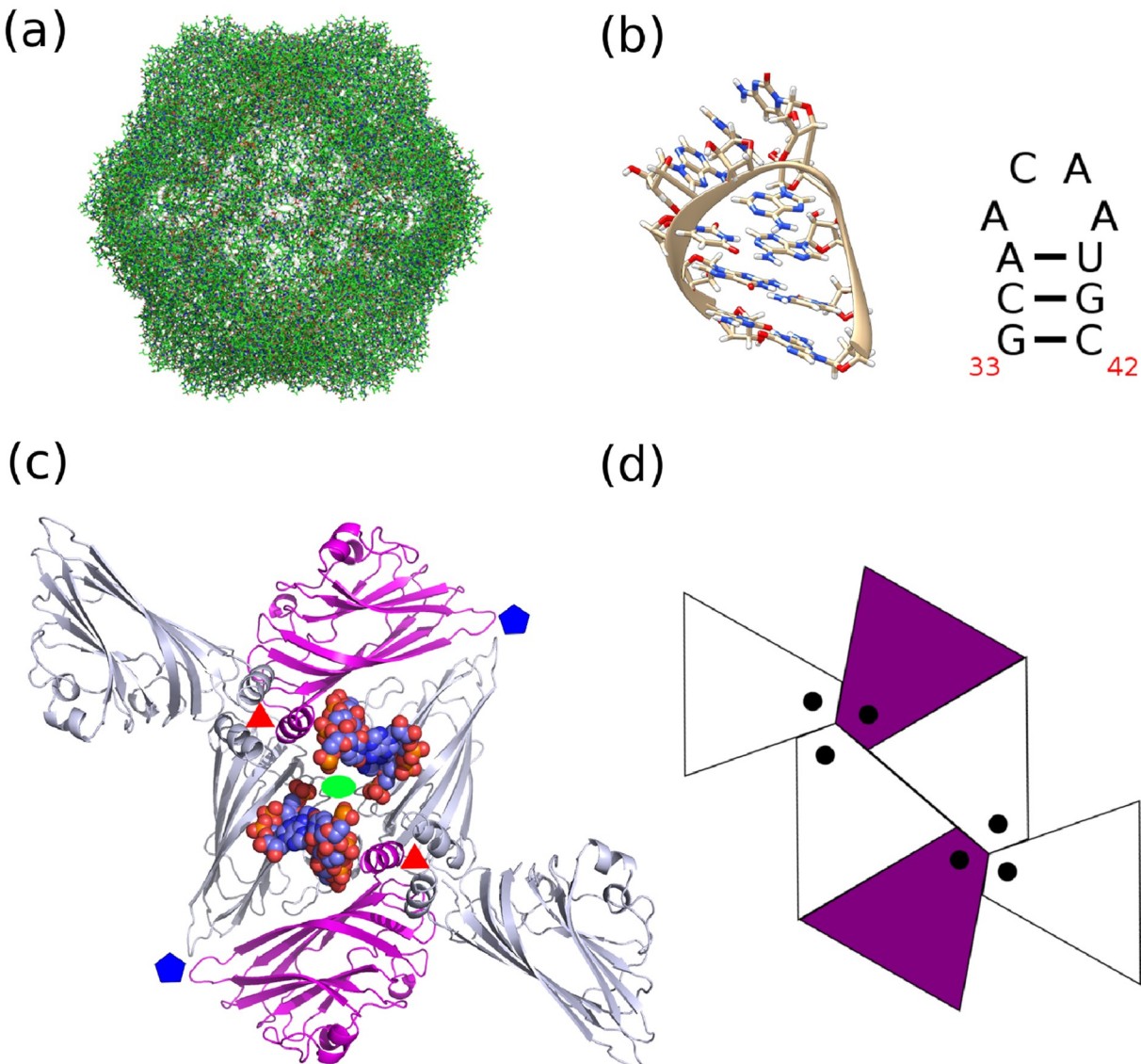

**Fig 1. Structure of the STNV capsid and orientation of RNA contacts.** (A) The $T = 1$ STNV capsid shell contains 60 copies of a 196 amino acid coat protein monomer in a beta jelly roll fold. (B) The predicted RNA fold of the B3 RNA hairpin identified by RNA SELEX [18]. (C) Orientation of the B3 RNA hairpin on the inner surface of the capsid as seen in [16] reveals that the backbone of the RNA hairpin contacts the N-terminal alpha helix domain of the protein at the three-fold symmetry axis of the capsid, while its apical loop extends towards the two-fold axis. (D) The two possible mutually exclusive orientations of the RNA helix (coloured purple in the net-diagram). We model the RNA as only being able to contact one of the N-terminal alpha helices, depicted by black dots in the purple shaded rhombs, when extending the apical loop towards the two-fold axis of the capsid.

determination resulted in 60 copies of RNA helix being observed. It is estimated from the X-ray density that roughly 50% of the 60 RNA segments are occupied, suggesting that approximately 30 RNA hairpins, (equivalently one per two-fold axis), contribute to the observed density. The orientation of the hairpin in the X-ray structure shows that the RNA helix is in contact with the N-terminal alpha helix of the CP, which sits in a trimeric cluster around the three-fold axis of the capsid (c.f. Fig 1c). The apical loop however is not observed, but the orientation of the A-form RNA helix suggests that the apical loop of the RNA is situated towards the two-fold axis of the capsid (c.f. Fig 1c). Thus, the observation of half filled RNA density

over 60 sites in the structure together with the structural data leads us to hypothesise that two orientations of the RNA hairpin exist, as illustrated in Fig 1d, where the RNA helix contacts the N-terminal alpha helix on the three-fold axis of the capsid and extends the apical loop of the hairpin towards the neighbouring two-fold. Thus, RNA occupying one of these orientations extends the apical loop of the hairpin towards the two-fold axis of the capsid shell, negating the possibility of a second hairpin occupying the alternative orientation, as this could cause a steric clash between apical loops. We term these two orientations as "mutually exclusive" and only allow one such site to be contacted during the assembly simulations of the capsid.

### Stochastic assembly model for ssRNA viruses with PSs

Previously, we developed a stochastic model for studying virus assembly around a ssRNA genome containing dispersed PSs [1, 10]. We briefly review the model here and discuss some minor alterations which were required to study assembly and the impact of local assembly rules in STNV. Fig 2a shows the two basic reactions of the STNV assembly process; (1) protein-RNA interactions which recruit new CPs to the partially assembled RNA/capsid complex, and (2) protein-protein interactions between CPs bound to neighbouring PS sites in the RNA. The assembly process begins with a nucleation step, where two CPs bound to adjacent PS sites in the RNA bind to each other to form a nucleation complex (c.f. Fig 2a). This nucleation step

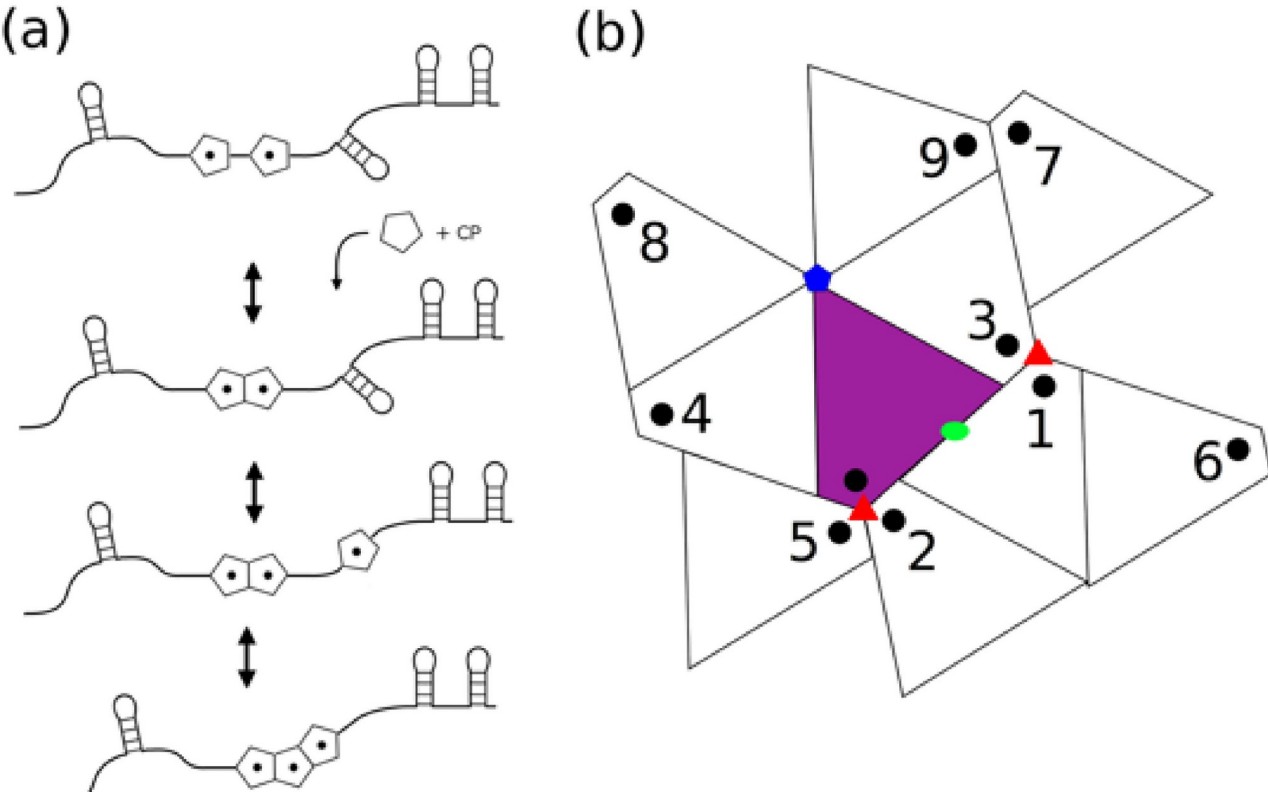

**Fig 2. Reactions in the assembly model of the STNV capsid.** (A) Two reaction types are considered in the assembly model, CP-PS binding and CP-CP binding. The nucleation step of the STNV capsid assembly occurs when two CPs bind and elongation of capsid assembly follows via attachment of CP: PS complexes to the growing capsid shell. The attachment of CP:PS complexes are controlled by the local assembly rules. (B) Local assembly rules governing the incorporation of CP:PS complexes into the growing capsid shell. The moves that the RNA path can potentially make from the central unit (indicated by purple shade) are numbered 1-9. The five-, three-, and two-fold symmetry axes of the capsid are indicated by the coloured pentagon, triangles, and oval.

is allowed to occur between any pair of two adjacent PSs bound to CP. Once nucleated, the capsid can grow by sequential binding of CP to the PSs 5′ or 3′ to the nucleated complex, followed by association of the CP:PS complex to the capsid shell (c.f. Fig 2a). We assume that CPs can reversibly bind to any section of the RNA containing PSs and that the lifetime of the CP bound to a PS will be determined by its affinity for this PS. Thus, our assembly model for STNV contains 30 PS sites within the RNA genome, each with a potentially different affinity for CPs. These affinity values are based both on an electrostatic component and contributions from sequence specific interactions. The difference in sequence specific interaction accounts for differences in affinity across the ensemble.

Recently, an analysis of the STNV capsid and the RNA contacts seen in Ford et al. [16] revealed potential ways in which the RNA strand could connect between neighbouring PSs in contact with the capsid [19]. Based on this analysis, we determine the local assembly rules i.e., the possible ways that a CP:PS complex that is 5′ (or 3′) to the growing capsid could associate to the partially formed capsid. We refer to these possibilities as a "move" and the potential moves are shown in Fig 2b. Starting at the contact shown in purple, we label all binding sites that are available as rotations about an axis that is flanking the protein. We note that contacts 3,4,8,9 are moves around a proteins' five-fold axis; 2,5 are moves around a proteins' closest three-fold axis; 6,7 are moves around a proteins' second closest three-fold axis; and 1 is a move around a proteins' closest two-fold axis. We refer to a subset of these as a "move set" and examine different move sets and their impact on assembly efficiency and RNA organisation. The choice of moves used determines the potential paths, which indicate genome organisation in proximity to the capsid. Due to the icosahedral symmetry of the capsid shell, if a move is included in a move set we also include the equivalent symmetry related move as well. For example, move numbers 3 and 4 in Fig 2b describe the RNA tracing out a path around the five-fold axis of the capsid and can be thought of as rotations by $\frac{2\pi}{5}$ about the five-fold axis, one clockwise and one anticlockwise. Thus, moves 3 and 4 in this case are inverses of each other from the perspective of the rotational operations that leave the STNV capsid invariant, with move 1 the only move which is self-inverse.

Stochastic simulation of the assembly reactions outlined in Fig 2a are performed using the Gillespie algorithm [20] with details specific to RNA virus assembly given in [10]. We make two additional modification to this algorithm to adapt it to STNV. Since there are potentially two orientations in which PSs can contact the two-fold axes of the capsid shell (c.f. Fig 1d), we exclude CP:PS complexes from being incorporated into positions on the capsid which would result in two RNA hairpins occupying mutually exclusive orientations as illustrated in Fig 1d. Second, in a previous bio-informatics study of the RNA folds of STNV [18, 21], we identified approximately 30 putative packaging signal sites within the STNV RNA which had similar fold and apical loop structure with the hairpin identified via SELEX (Fig 1b). Thus, given the bio-informatics study, combined with the structural data suggesting there are approximately 30 RNA hairpins present, we consider the RNA to contain 30 PSs contacting binding sites at one of the 30 two-fold axes of the capsid during assembly.

## Graph theory and hamiltonian paths description of RNA organisation

The association of CP:PS complexes to the growing capsid under the local assembly rules outlined by the possible moves shown in Fig 2b results in an RNA organisation in proximity of the capsid shell that can be modelled as a path connecting the 30 PS contacting the 60 CP subunits which make up the capsid. Thus, for each fully assembled capsid in our simulation, the RNA will trace out a path (5′ to 3′ on the RNA) on the inside surface of the capsid, and our assembly program outputs this information as an ordering of the 60 protein subunits which make up the capsid. By expressing the capsid as a graph with 60 vertices (one vertex at each CP

position), these RNA paths can be represented as a series of edges connecting pairs of vertices in the graph, resulting in a connected graph in which exactly 30 of the vertices are contacted. Each edge represents a move that the RNA can make between CP binding sites. More importantly, if one was to move along the path of edges starting from the vertex representing the first PS in the RNA (at the 5′ end) to the last PS in the RNA (at the 3′ end), 30 vertices would be visited only once and 30 vertices would not be visited at all. The 30 non-contacted vertices are thus a consequence of the two mutually exclusive orientations of the RNA in contact with the two-fold axis that we hypothesise based on the X-ray structure [15] (c.f. Fig 1d).

In previous assembly models with MS2 [22] and a simplified dodecahedron model of picornavirus assembly [10], the paths that represent RNA organisation at the inner shell of the capsid are Hamiltonian paths, i.e. a sequential path of edges through the graph of all possible assembly pathways (the assembly graph), where every vertex in the graph corresponds to a binding site and is visited exactly once. Due to the mutual exclusivity criterion, the RNA paths do not visit every vertex here, but they are guaranteed to visit at least one in each mutually exclusive pair. Given that every pair is visited precisely once, we call the RNA paths in STNV pseudo-Hamiltonian paths. More formally, they are defined as follows: consider all sub-graphs of the assembly graph that include exactly one vertex from each mutual exclusive pair of vertices. Then a Hamiltonian path on these sub-graphs is called a pseudo-Hamiltonian path on the assembly graph of the capsid. Here, our RNA paths only visit a subset of 30 out of 60 vertices, and hence these are not Hamiltonian paths from the standpoint of the full graph containing 60 vertices. However, if we group the vertices together into a set of 30 mutually exclusive pairs, then the RNA paths in the STNV assembly model visit each mutually exclusive pair of vertices precisely once, akin to a proper Hamiltonian path. Thus, we refer to the RNA path through 30 pairs of mutually exclusive vertices as a pseudo-Hamiltonian path. Fig 3 shows the sixty vertices (30 pairs) with an example pseudo-Hamiltonian path for the move set {2, 3, 4, 5}. The combinatorics of the number of pseudo-Hamiltonian paths and their connectivity will depend on the move set encoding the allowed transitions of the RNA between vertices in the graph. These pseudo-Hamiltonian paths are useful in describing the sequential order in which contacts between RNA and protein shell are made, and thus provide information on RNA organisation in proximity to the inner capsid surface. As such, they provide constraints on the overall RNA organisation that could be used in combination with constraints derived with different techniques by other authors [23], but are not sufficient to fully characterise RNA organisation within the capsid.

## Optimisation of PS affinities for efficient assembly

Given a move set and an RNA containing 30 PSs with varying affinities for CP, the stochastic assembly algorithm can be utilised to determine the average number of capsids that correctly assemble at thermodynamic equilibrium. Since our RNA virus assembly model allows for each of the 30 PSs to have different affinities for CP, and thus a different average lifetime in which the CP:PS complex will exist for [10], we can investigate the impact that a heterogeneous distribution of PS affinities in the RNA has on the efficient assembly of STNV capsids. For a given move set, we utilise a genetic algorithm to systematically adjust the affinities of the PSs in the RNA and identify a variant with high yield of correctly assembled capsid in thermodynamic equilibrium.

The genetic algorithm is outlined as follows. Starting with 1024 RNA variants, each with a random distribution of PS-CP binding free energies distributed between −4 and −12 kcal $M^{-1}$, we perform an STNV assembly simulation on a homogeneous population of 2000 copies of each RNA variant and determine the maximum number of the 2000 RNAs that are fully

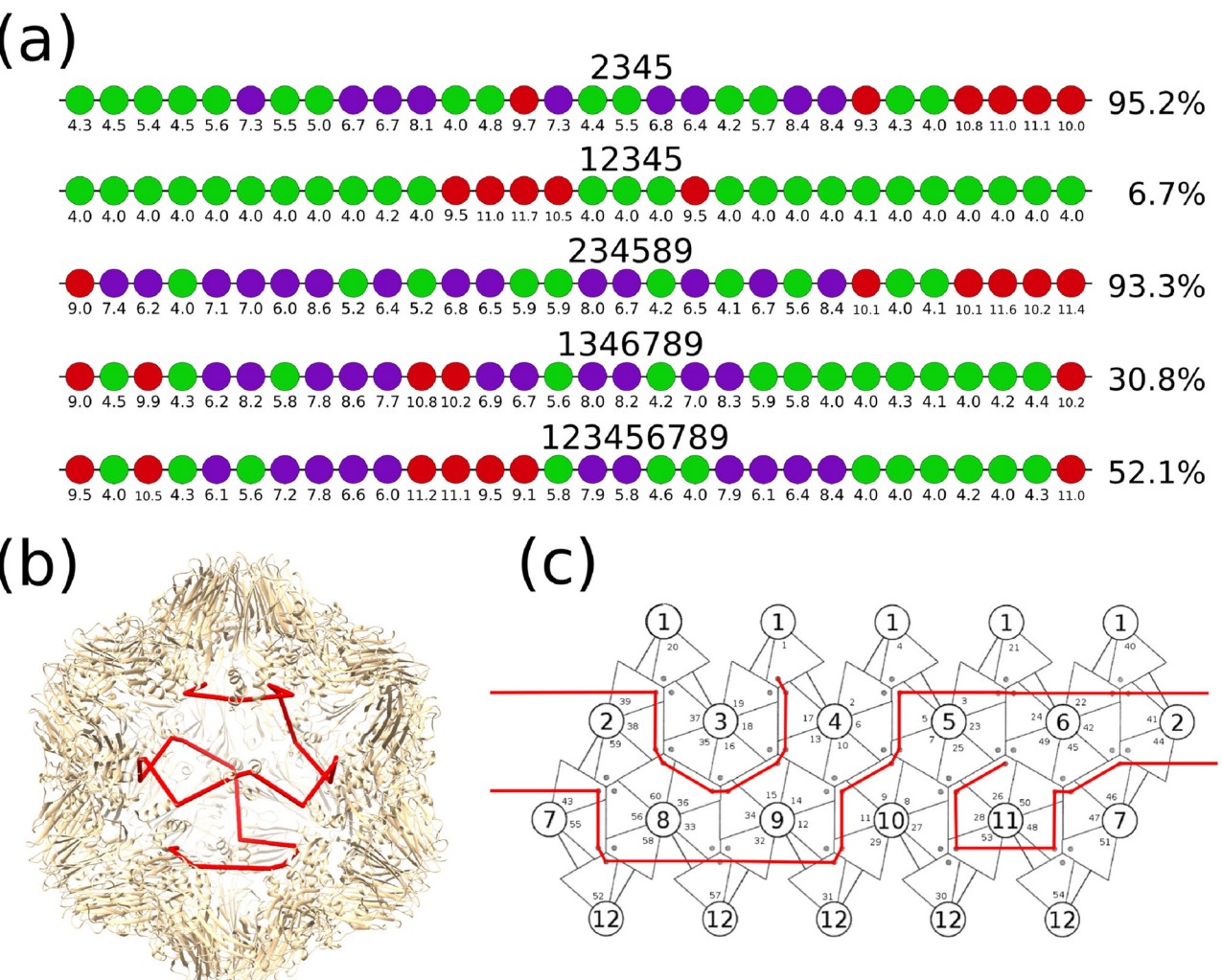

**Fig 3. The optimal PS-CP binding free energies and the efficient assembly path.** (A) The results from the optimisation of the PS-CP binding free energies $\Delta G_{ps}$ for each move set with corresponding average percentage capsid assembled. PS-CP binding energies are colour coded according to the bands −4.0 to −5.9 kcal M$^{-1}$ (green), −6.0 to −8.9 kcal M$^{-1}$ (purple), and −9.0 to −12.0 kcal M$^{-1}$ (red). (B) The crystal structure of the STNV capsid, composed of 60 protein subunits (PDB 4v4m), is shown together with an example path (red) which indicates the order in which contacts with the protein shell are made. (C) A schematic representation of the capsid surface as a tiling, with numbering of protein subunits (small) and five-fold symmetry axes (large). An example path, corresponding to the path in (B), is shown. This path corresponds to the most frequently occurring path for the move sets {2, 3, 4, 5} and {2, 3, 4, 5, 8, 9}.

packaged into assembled capsid at equilibrium. The top 256 RNA variants (out of 1024) with the highest quantity of assembled capsids are retained, and each of their 30 PSs are mutated (with a probability of 5% for each PS) to create 768 new RNAs and the next generation of 1024 RNA variants. The algorithm continues for several generations until the top 20 RNA variants have very similar yields, and there is no new RNA variants joining the top 20. After 20-30 generations, the top 20 RNA variants all had similar yield of fully assembled capsids, and there is little to no further improvement in subsequent generations. The position of high affinity PSs are significant to the outcome of assembly. For the first generation of RNA variants, the number of high affinity PSs, at each position along the RNA and in total on the RNA, were calculated. In the first generation, each position was equally likely to have a strong affinity PS (-8 to -12 kcal/M) vs a low affinity one (-4 to -8 kcal/M) and the number of strong affinity PSs in

each RNA was normally distributed. This suggests that we have interrogated the assembly landscape using a representative set of RNA variants having strong PSs in all positions along the RNA. Moreover, starting with different random distributions of PSs in the first generation does not identify other local maxima in the fitness landscape with higher assembly yield. These results are similar to what was seen previously in an assembly fitness landscape on an explicit RNA sequence where the yield of assembled capsids plateaus after a few generations, suggesting the existence of multiple local maxima in the assembly fitness landscape with near identical fitness [24]. The assembly software along with the initial 1024 RNAs used for generation 1 can be downloaded from https://github.com/edykeman/stnvassembly or requested from the author via email.

## Results

### Combinatorial analysis of STNV RNA paths

We construct all possible pseudo-Hamiltonian paths which represent a potential RNA organisation for the inner surface of the STNV capsid for different move sets. We consider move sets containing between 4 and 9 total moves encompassing all move sets which are possible. To determine the number of pseudo-Hamiltonian paths for a given move set, we developed a basic depth-first recursive search algorithm to generate, count, and analyse all possible unique pseudo-Hamiltonian paths. All combinations of moves that could generate a move set containing 4-9 moves were constructed, using our depth-first search algorithm to construct an RNA path and keep track of the total number of unique paths that can be constructed. The number of unique paths for each move set studied are detailed in Table 1. While the move set {3, 4, 8, 9} is potentially possible since it contains moves 3 and 8 along with their inverses, it was excluded as it cannot create any pseudo-Hamiltonian paths and only describes movement of the RNA around a single five-fold axis of the capsid.

Table 1 details the combinatoric analysis of the number of unique pseudo-Hamiltonian paths for each move set. As can be seen from the table, larger move sets contain (in general) more possible paths, with some combination of moves in a move set (e.g. using moves 8 and 9 instead of moves 3 and 4) increasing the number of path options. However, move set {2, 3, 4, 5} seems to be exceptional overall, having an unusually small number of possible paths when compared to other move sets, as well as other move sets with 4 moves. Moreover, move sets which contain other move sets as a subset also contain all of the paths for the subset as a solution. For example, the larger move sets {1, 2, 3, 4, 5}, which contains move set {2, 3, 4, 5} as a subset, has the 64 unique paths from move set {2, 3, 4, 5} along with 120024 additional paths which must contain the additional move number 1 somewhere in the path. This reveals the complexity that adding a single additional move can add regarding the number of path options during assembly.

### Assembly efficiency of different move sets

Given the different move sets, we computationally assessed each move set's ability to form complete capsids using our stochastic assembly model outlined in Methods, which takes into account the mutual exclusion principle in addition to the standard assembly model detailed [1, 10]. For each move set, we simulated assembly of 2000 RNAs, each containing 30 identical PSs with a PS-CP binding energy set to $\Delta G_{ps} = -9$ kcal $M^{-1}$, with sufficient coat protein to fully assemble all 2000 RNAs into complete capsids. The protein-protein binding energy for contacts between neighbouring proteins was set to $\Delta G_{cs} = -4.0$ kcal $M^{-1}$. Simulations were run until thermodynamic equilibrium, and the total number of assembled capsids at this point was rounded to the nearest 10. Table 1 details the results of the assembly simulations for each

**Table 1. Combinatorial and assembly analysis of different STNV move sets.** For each of the different move sets, we computed the number of unique pseudo-Hamiltonian paths ($N_{path}$) and performed a stochastic assembly simulation with 2000 copies of an RNA containing 30 PSs with identical affinities (CP-PS binding energy set to $\Delta G_{ps} = -9$ kcal $M^{-1}$) and determined the number of capsids assembled at equilibrium ($N_{cap}$). Data for the number of pseudo-Hamiltonian paths ($N_{path}$) was obtained using our in house depth-first search algorithm. The number of capsids assembled was rounded to the nearest 10 and move sets which generated a negligible but non-zero number of capsids have been marked as "<10" in the table. *The total number of unique paths for move sets with 8 or 9 moves was not calculated as it was computationally prohibitive.

| N | Move set | $N_{cap}$ | $N_{path}$ |
|---|---|---|---|
| 4 | 2345 | 760 | 64 |
| 4 | 2567 | 0 | 1342 |
| 4 | 2589 | 0 | 12582 |
| 4 | 3467 | <10 | 826 |
| 4 | 6789 | 0 | 7242 |
| 5 | 12345 | <10 | 120086 |
| 5 | 12567 | 40 | 12937410 |
| 5 | 12589 | 20 | 26672730 |
| 5 | 13467 | 80 | 2240200 |
| 5 | 13489 | <10 | 3510708 |
| 5 | 16789 | 30 | 12965858 |
| 6 | 234567 | 160 | 2959693816 |
| 6 | 234589 | 640 | 2117037620 |
| 6 | 256789 | 0 | 4399719160 |
| 6 | 346789 | 10 | 2501657142 |
| 7 | 1234567 | 260 | 279417463096 |
| 7 | 1234589 | 110 | 362090241596 |
| 7 | 1256789 | 180 | 856244868982 |
| 7 | 1346789 | 360 | 524658774154 |
| 8 | 23456789 | 300 | * |
| 9 | 123456789 | 730 | * |

move set. Some move sets are unable to form any assembled capsids. Specifically, the data shows that a move set must include either move {1} or moves {3, 4} to have any assembled capsids. This is because a new CP can only be incorporated into the existing partial capsid structure if the incoming CP binds to one of the existing CPs in the structure, which move {1} and moves {3,4} allow. Thus, having at least one of these three moves present in a move set is required to allow the formation of protein-protein interactions between incoming CPs and the partial capsid structure. From a group theory perspective, the inclusion of either moves {1} or moves {3,4} with moves {2,5} ensures the move set encompasses a generating set of the icosahedral group (e.g. a five-fold and three-fold rotation).

Moreover as can be seen in Table 1, increasing the number of overall moves in the move set results, in general, in more efficient capsid assembly, presumably due to an overall expansion of the number of assembly paths. Interestingly and somewhat unexpected however, the best performing move set was {2,3,4,5} with the smallest number of assembly paths. In addition, of the three move sets with the highest number of assembled capsids, the moves {2, 3, 4, 5} were always present, suggesting that this core set of moves enables efficient assembly to occur. When adding move 1 to a move set that already included {2,3,4,5}, often the number of assembled capsids decreased, while adding it to other move sets tended to increase the number of assembled capsids, though usually only by a slight amount. However, although it appears that move set {2,3,4,5} outperforms other move sets under conditions where the RNAs have a

uniform distribution of PS-CP binding free energies, it might be possible that alternative move sets are better able to optimise their PSs for efficient assembly, i.e. the fitness landscape may be more optimal in an alternative move set. We explore this possibility in the next section.

## Identification of RNAs with PS distributions optimised for assembly

For any of the move sets listed in Table 1, the genetic algorithm for PS optimisation detailed in Methods can be used to identify an RNA with a PS configuration that is ideal for efficient assembly. When optimising the 30 PS-CP binding free energies in an RNA for a particular move set, we consider the "assembly fitness landscape", i.e. the hyper-surface which maps a 30 dimensional coordinate of PS-CP binding free energies to a value of the percentage capsid assembled. Since the preliminary data in Table 1 highlights that move set {2,3,4,5} is critical, we have selected five move sets to optimise the PS distributions for and explore their assembly fitness landscapes. We have chosen to optimise the PS distribution for RNAs under the following move sets {2,3,4,5}; {2,3,4,5,8,9}; {1,2,3,4,5,6,7,8,9}; {1,2,3,4,5}; and {1,3,4,6,7,8,9}. The first three move sets were chosen as they were the ones with the highest numbers of capsid assembled under the condition of uniform PS-CP binding free energies. The last two move sets were chosen to explore; (1) the role of the move subset 2,3,4,5 and any negative effects of move 1 on assembly efficiency, and (2) if the assembly fitness landscape for move sets such as {1,2,3,4,5} actually contains a highly optimal RNA variant which outperforms other move sets such as {2,3,4,5}. Due to its relatively high number of assembled capsid and the lack of {2,3,4,5} as a subset of moves, the move set {1,3,4,6,7,8,9} was also included.

These five move sets were each optimised following the genetic algorithm detailed in Methods, and the optimal RNA for each move set is shown in colour-coded dot representation in Fig 3a along with the average percentage of fully assembled capsid at thermodynamic equilibrium. In addition to this, we have also calculated the average percentage of assembled capsid for uniform PS distributions with binding free energies of $\Delta G_{ps}$ = -4, -8, and -10 kcal M$^{-1}$. The average percentage of assembled capsid was computed from an average of 5 stochastic simulations. Table 2 highlights the data for each of the five move sets that we explored.

In almost all cases, the percentage of capsid assembled increased noticeably through the optimisation of PS-CP binding free energies on the RNA. For move set {1,2,3,4,5} however, variation of PS-CP binding free energies was unable to improve the percentage of capsid assembled to higher than 7%, confirming that the addition of move 1 to the move set is unfavourable to the assembly kinetics. Interestingly, although move sets {2,3,4,5} and {1,2,3,4,5,6,7,8,9} have similar assembly efficiency for RNAs with uniform PS distributions, the genetic algorithm reveals that the assembly fitness landscape is substantially different, with the genetic algorithm only able to identify an RNA variant able to assemble 52% of RNAs into

**Table 2. Comparison of assembly efficiency for RNAs with optimised and uniform PS distributions.** For each move set, the average percentage of assembled capsid was computed from 5 stochastic simulations using an RNA with one of the four PS distributions; uniform $\Delta G_{ps}$ = −4, −8, −10 kcal M$^{-1}$, or the optimal RNA PS distribution identified from the genetic algorithm. Additionally, two other PS distributions were tested. The first (knockout) involved taking the optimal RNA for a move set and setting the nucleation site, i.e. the section of 2-4 PSs with high affinity for CP shown as red dots in Fig 3, to $\Delta G_{ps}$ = −4 kcal M$^{-1}$. The second (Nuc. Only) set all other PSs, except the nucleation site, to $\Delta G_{p}$ $s$ = −4 kcal M$^{-1}$.

| Move set | -4 | -8 | -10 | Optimal | Knockout | Nuc. Only |
|:---:|:---:|:---:|:---:|:---:|:---:|:---:|
| 2345 | 30.72% | 35.68% | 34.93% | 95.20% | 3.88% | 93.43% |
| 12345 | 2.28% | 0.08% | 0.15% | 6.69% | 0.26% | 4.23% |
| 234589 | 22.14% | 32.88% | 31.62% | 93.33% | 9.62% | 65.73% |
| 1346789 | 12.00% | 17.57% | 17.86% | 30.77% | 21.26% | 14.38% |
| 123456789 | 28.24% | 34.04% | 35.17% | 52.06% | 37.86% | 30.62% |

capsids. This is similar with the move set with seven moves {1,3,4,6,7,8,9} which appears to have an essentially flat assembly fitness landscape. This leaves just two move sets, {2,3,4,5} and {2,3,4,5,8,9}, which give very high percentages of assembled capsids when the PS-CP binding free energies on the RNA are fully optimised.

Further examination of the PS distributions for the five move sets reveals that a group of 2-4 high-affinity PSs (low PS-CP binding free energy) occupies either the right most end of the RNA (in move sets {2,3,4,5} and {2,3,4,5,8,9}) or is positioned roughly in the middle of the RNA (in the remaining move sets). We have previously noted that the presence of an area of 2-4 co-located high affinity PSs in the RNA is critical for nucleation [10] and selective packaging over host mRNAs [1]. These main features are found again here, but interestingly, the geometry of the local assembly rules (i.e. which move set is used) determines the optimal position of the nucleation region containing the 2-4 high affinity PSs. Previous experimental work done by Patel and colleagues [2] has shown that altering of putative PS sites in the first 120 nucleotides of the 5' end of STNV RNA results in an RNA variant with poor assembly *in vitro*. We find a similar result in our model when we knockout the nucleation sites in the optimal RNAs (column Knockout in Table 2) and replace these high affinity sites with low affinity PSs.

Our modelling results indicate that the assembly fitness landscapes for these move sets substantially differ as a result of the local geometry rules utilised during capsid assembly. Since the assembly fitness landscape is a 30-dimensional hyper-surface, explicit exploration and plotting of the surface of the landscape for any of the move sets is not feasible. However, Fig 4 shows histogram plots of the percentage of capsids assembled for the initial population of 1024 random RNA variants over four of the move sets with $\geq 20\%$ capsid assembled at equilibrium. The histogram plots give a rough indication of the overall shape of the assembly fitness landscape and the probability of finding an RNA capable of assembling a high percentage of its RNAs into fully formed capsids. Interestingly, while move sets {2,3,4,5} and {2,3,4,5,8,9} (Fig 4a and b, respectively) have a majority of their landscape flat and in the range of 5-30% capsid assembled, they have long thin "tails" at the extremes suggesting only a few options of optimal local maxima exist in the assembly fitness landscape. In contrast, move sets {1,3,4,6,7,8,9} and {1,2,3,4,5,6,7,8,9} (Fig 4c and d, respectively) have a very flat landscape with no solutions of PS distributions which have either high or low assembly yield. This suggests that these move sets are predominately neutral to changes in PS affinities and that any arbitrary RNA would achieve similar assembly yields compared with any other.

## Analysis of preferred RNA paths in assembled capsids

Since our stochastic assembly program can track and output the exact path used by the RNA during capsid assembly, we have used the resulting paths from the assembly data from the 5 stochastic simulations to compute the frequency of specific pseudo-Hamiltonian paths in the ensemble. When the pseudo-Hamiltonian paths of the optimal RNA from each move set were analysed, some unexpected results were found. Neither move set {1,3,4,6,7,8,9} nor {1,2,3,4,5,6,7,8,9} identified any paths that were repeated more then five times over the 3k-5.2k paths that were generated from the five stochastic simulations. This is likely to be due the amount of available assembly options and overall paths for these larger move sets. In contrast, the three smaller move sets {2,3,4,5}, {1,2,3,4,5} and {2,3,4,5,8,9} all exhibited a strong bias towards a small number of specific assembly paths, as shown in Fig 5. This could be due to the fact that these move sets contain a smaller number of moves than the other sets. Whilst the sample of paths obtained from move set {1,2,3,4,5} are very small (less then 800 paths), there appears to be a preference for several path types, similar to move set {2,3,4,5,8,9}, and the paths

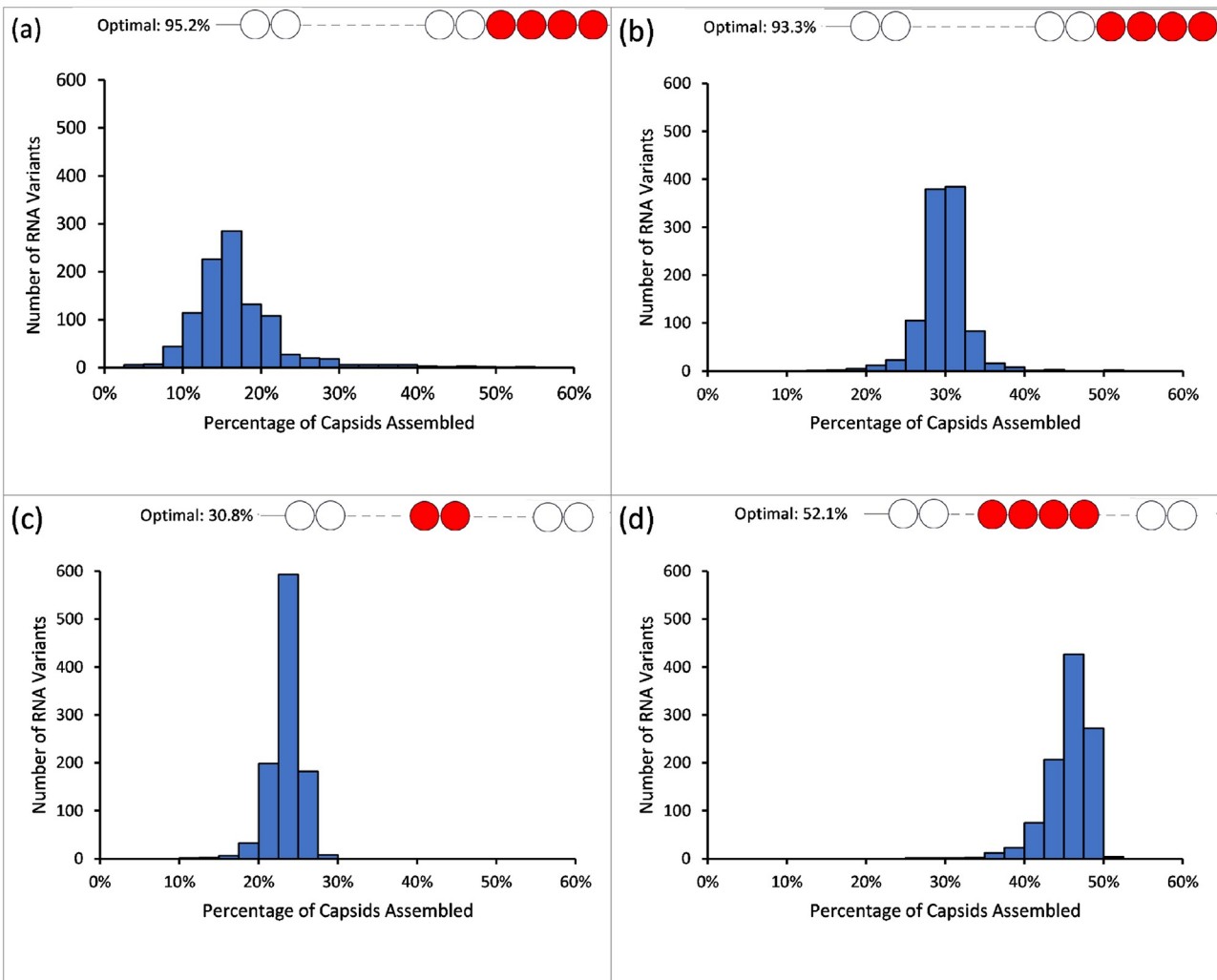

**Fig 4. Histogram representations of assembly fitness landscapes.** A histogram representation of the 30-dimensional assembly fitness landscape for four different move sets. A set of 1024 RNA variants with random $\Delta G_{ps}$ were selected and the number of assembled capsids for each were computed in separate stochastic assembly simulations. The percentage capsid assembled for each of the 1024 RNAs was computed and plotted as a histogram for move sets (A) {2,3,4,5}, (B) {2,3,4,5,8,9}, (C) {1,3,4,6,7,8,9}, and (D) {1,2,3,4,5,6,7,8,9}. The inset shows the position of the position of the high affinity PSs in the RNA corresponding to the fitness peak.

tended to differ mostly in the six moves near the centre of the path. However, due to the small sample size, definitive conclusions cannot be drawn for this move set.

Similarly, both of the move sets with high assembly efficiency, {2,3,4,5} and {2,3,4,5,8,9}, had a very strong preference for certain paths based on a sample of 9520 paths and 9333 paths, respectively, taken from fully assembled capsids. For move set {2,3,4,5}, there are two path types (P and P*) that make up 90% and 9%, respectively, of the 9520 total paths from our sample set (c.f. Fig 5). These two paths are highly similar, being identical up until the four moves at the end of the RNA. The difference between these two path types amounts to whether the RNA goes around the last five-fold axis in the capsid to assemble either clockwise (path P) or anticlockwise (path P*). Move set {2,3,4,5,8,9} on the other hand has a slightly larger variety of paths. When the sample of 9333 paths were analysed, 9227 (98.8%) were identical with paths P and P* up until the four moves at the end of the RNA (c.f. Fig 5). As there are an increased number of ways to complete these four moves for the move set {2,3,4,5,8,9} when compared

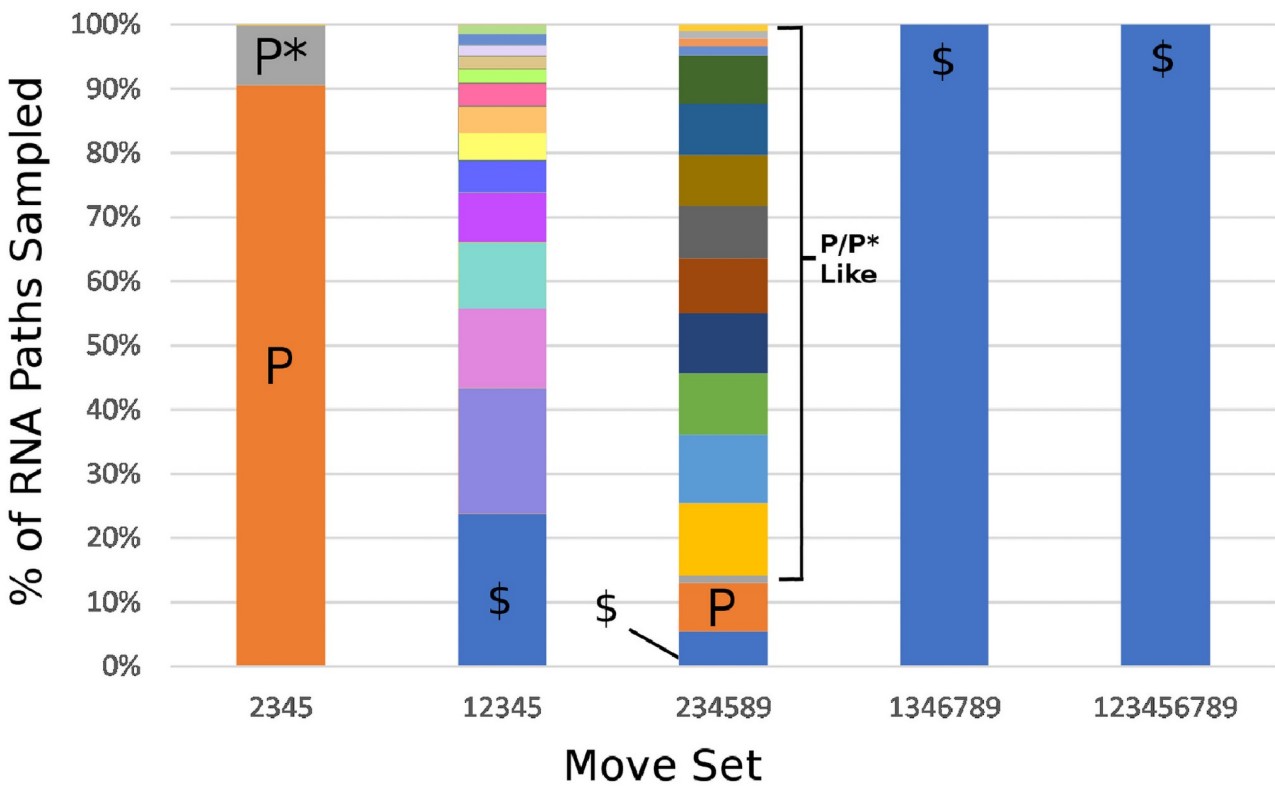

**Fig 5. Relative frequencies of different path types in assembled capsids.** Assembled capsids from move sets {2,3,4,5} resulted in two predominant paths (P and P*) which make up over 99% of the fully assembled capsids. Path P is illustrated in Fig 3a. Move set {2,3,4,5,8,9} resulted in a variety of P and P*-like paths which only differ in the four moves at one end of the RNA. These variations in the four moves at the end of the RNA corresponded to different nucleation scenarios on the first five-fold axis of the capsid. The symbol $ is used to denote paths which are sampled ≈5/10000 times in the ensemble of sampled paths.

with {2,3,4,5}, this resulted in less specificity on the assembled paths and we identified 30 paths out of the $>2^{12}$ (c.f. Table 1) which were identical with paths P and P* up until the four moves at the end of the path. Thus, despite being given two extra moves to use in assembly (moves 8 and 9), the paths utilised by the RNA during capsid assembly essentially followed the same paths P and P* used by the smaller move set {2,3,4,5}. This was despite having over 2 trillion assembly paths versus just 64 for move set {2,3,4,5}. Note that despite move sets {2,3,4,5} having the smallest number of potential paths of any move sets, it generated the highest yield post optimisation.

The common RNA path (path P) for these move sets is illustrated on the inner surface of the capsids (Fig 3b) and as a planar-2D net (Fig 3c). The geometry of this path results in a simple spiral of the RNA around the interior surface of the capsid implying that the two hemispheres of the capsid are constructed sequentially. The four moves at the end of the RNA, which distinguish between paths, corresponded to the positions of the four high-affinity PS:CP contacts representing the nucleation site in the RNA (c.f. Fig 3a). Thus, all of the variation in the RNA paths arises from the nucleation step and the formation of the first five-fold axis of the capsid. Whilst using moves {2,3,4,5}, once the five-fold axis has been assembled, there is only one path for the RNA that leads to completion of the capsid. This is because with only these four moves, there are a lot of restrictions on what moves can be selected at a given point: it cannot go backwards to the same position a second time, so all steps but the first have only 3 choices. Additionally, the mutual exclusion principle prevents the RNA from doing move 2 after 4 or

move 3 after 5, which limits choices further. The same effect is shown when nucleating the capsid by occupying the three-fold axis in the same way. This makes it clear that formation of the nucleated five-fold in this case is critical to ensuring a high probability of completing capsid assembly. Finally, to test the importance of the nucleation site and the role of the remaining PSs in the RNA, we performed additional assembly simulations in which either (1) the four high affinity PSs where knocked out (Table 2—Knockout), or (2) where the other PSs where set to low affinity (Table 2—Nuc. Only). From the knockout simulations we identified a strong reduction in the percentage of capsid assembled, performing even worse than the uniform PS distributions. However, for the move set {1,2,3,4,5,6,7,8,9} the knockout variant had only a mild reduction in the percentage of capsid assembled. From the nucleation only simulations, we found that there was only a mild effect on the capsid yield in move set {2,3,4,5}, but a much more dramatic result in move set {2,3,4,5,8,9}.

## Discussion

Assembly in ssRNA viruses involves a cooperative co-assembly process in which capsid proteins and ssRNA assemble spontaneously into a final capsid structure. Ensuring efficient assembly of progeny virus containing viral RNA over host mRNA is a critical issue for this class of viruses. This has led to two main hypotheses for how assembly takes place in these viruses. The first, a PS mediated assembly model, postulates that multiple PS sites contribute to efficient assembly [25] and selectivity over host mRNAs occurs via nucleation at high affinity PS sites that are present in viral, but not host mRNA. The second assumes that capsid assembly around RNA occurs via largely non-specific interactions such as electrostatic charge neutralisation, and the resulting RNA organisation in the capsids is largely disordered [4, 9, 26]. Selective packaging of viral over host mRNAs could occur either via viral assembly "factories" where viral mRNAs are predominantly present in areas of the cell where assembly is actively taking place, or via a single packaging signal site, which would allow for the formation of an assembly competent nucleus with remaining packaging of the RNA into the capsid controlled largely though non-specific interactions.

In bacteriophages MS2 and Q$\beta$, there is now a variety of experimental [3] and theoretical [27] evidence that packaging occurs via multiple packaging signal sites, with asymmetric cryo-EM reconstructions now visualising specific RNA hairpins in contact with capsid proteins on the inner capsid surface [3]. However, in BMV and CCMV, recent cryo-EM reconstructions have observed largely disordered RNA with no specific RNA contact observed [4]. Although BMV and CCMV have been shown to assemble via mostly non-specific interactions *in vitro* [9, 26], there is evidence that tRNA structures in the viral RNA may be important for selective packaging *in vivo* [28]. Therefore it is difficult to determine the extent to which PSs may play active roles during assembly in viruses such as CCMV and BMV.

Previously we have demonstrated using our assembly model how high affinity nucleation sites present in viral, but not host mRNA, can lead to selective packaging in ssRNA viruses [1]. In this work, we have explored how the local assembly rules for the capsid shell can affect assembly efficiency, with a specific application to the satellite plant virus STNV. Interestingly, we have identified scenarios for the local assembly rule which not only impact on the number of capsids assembled, but also on the placement of the nucleation site in the RNA (either 5$'$ / 3$'$, or centrally located). Moreover, some of the local assembly rules resulted in the RNA packaging signal sites and their affinity for coat protein only having a very weak effect on the overall yield of correctly assembled capsids. The resulting assembly fitness landscapes for these local assembly rules (specifically {1,3,4,6,7,8,9} and {1,2,3,4,5,6,7,8,9}) seem to be largely flat with a narrow range of potential capsid yields regardless of the RNA packaging

signal profiles (c.f. Fig 4c and 4d and the number of capsids assembled for the optimal RNA vs heterogeneous RNAs in Table 2). Although there was still a preference for a nucleation site for these local assembly rules, knockout of the nucleation site had little effect on assembly (c.f. Table 2—knockout), and there was no preference for any specific RNA organisation in the assembled viruses.

This is in contrast to the local assembly rules governed by move sets {2,3,4,5} and {2,3,4,5,8,9}, which had a very high dependence on the packaging signal distribution and on the presence of a nucleation site. Knockout of the nucleation site at the end of the RNA (Table 2—Knockout), or knockout of the PSs involved in assembly post-nucleation (Table 2—Nuc. Only), resulted in only a small amount of capsid assembled (<10% and <66%, respectively, for move set {2,3,4,5,8,9}). Our modelling observations are consistent with the equivalent experimental knockouts for the STNV virus performed by Patel and colleagues which showed similar effects due to removal of nucleation PSs or PSs involved in assembly post-nucleation [2]. Moreover, the resulting RNA organisations of the assembled capsids produced by these local assembly rules are highly conserved, consistent with the highly conserved organisation seen in the bacteriophage MS2 RNA which also packages using multiple PS sites within its RNA genome. For move set {2,3,4,5}, this specificity is partly due to the fact that after the first five proteins have formed around the five-fold axis, there is only one way in which the capsid can be completed by the RNA. This can be interpreted as nature evolving restrictions that limit itself to ensure it gets the desired result.

The results from our assembly model give a possible explanation for the differing observations of packaging signals and their effects on assembly both *in vitro* and *in vivo*, as they suggest the potential for two solutions to the assembly problem in ssRNA viruses. The first solution is a packaging signal mediated model which relies on multiple PS sites that are highly evolved and specifically distributed throughout the viral RNA. In this case, our model predicts that there is a high efficiency for viral assembly when the viral RNA is "optimised" via a specific distribution of PSs (>90%) and a tendency to organise the viral genome in a highly conserved structure in the capsid. Our local assembly rules model for move sets {2,3,4,5} and {2,3,4,5,8,9} are consistent with a packaging signal-mediated model of capsid assembly, corroborating experimental observations in STNV and MS2. The second solution suggested by our model assembles capsids largely non-specifically (as the affinity of RNA-CP interaction sites is mostly irrelevant for assembly yield), but has a preference for a single nucleation site. The RNA organisation is mostly disordered in these capsids and the overall assembly efficiency is up to 50% for the $T = 1$ capsid geometry example used here. Our local assembly model for move sets {1,3,4,6,7,8,9} and {1,2,3,4,5,6,7,8,9} is consistent with this second type of assembly scenario and this could reflect the situation for viruses which do not seem to exhibit PS-mediated assembly. Thus, the local assembly rules of the capsid subunits may play a more crucial role in governing the overall assembly strategy (PS-mediated vs. non-specific) of ssRNA viruses. Future assembly experiments may need to consider if alteration of the chemical conditions *in vitro* may impact the local assembly rules of the protein subunits, thereby artificially switching between these two assembly strategies.

## Author Contributions

**Conceptualization:** Reidun Twarock, Eric C. Dykeman.

**Data curation:** Sam R. Hill.

**Formal analysis:** Sam R. Hill, Eric C. Dykeman.

**Funding acquisition:** Reidun Twarock.

**Investigation:** Sam R. Hill, Eric C. Dykeman.

**Methodology:** Reidun Twarock, Eric C. Dykeman.

**Project administration:** Eric C. Dykeman.

**Software:** Sam R. Hill, Eric C. Dykeman.

**Supervision:** Reidun Twarock, Eric C. Dykeman.

**Validation:** Sam R. Hill.

**Writing – original draft:** Sam R. Hill, Eric C. Dykeman.

**Writing – review & editing:** Sam R. Hill, Reidun Twarock, Eric C. Dykeman.

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
