## [Decision Letter · Decision Letter 0]

12 May 2021

Dear Dr. Dykeman,

Thank you very much for submitting your manuscript "The impact of local assembly rules on RNA packaging in a T=1 satellite plant virus" for consideration at PLOS Computational Biology. As with all papers reviewed by the journal, your manuscript was reviewed by members of the editorial board and by several independent reviewers. The reviewers appreciated the attention to an important topic. Based on the reviews, we are likely to accept this manuscript for publication, providing that you modify the manuscript according to the review recommendations.

Sincerely,

Shi-Jie Chen

Associate Editor

PLOS Computational Biology

Ilya Ioshikhes

Deputy Editor

PLOS Computational Biology

[LINK]

Reviewer's Responses to Questions

**Comments to the Authors:**

Reviewer #1: The authors have previously developed a stochastic model to investigate the assembly of capsid proteins around a ssRNA containing dispersed packaging signals. In this paper, they extend their previous work and consider that RNA has 30 packaging signals and investigate different pathways it connects the capsid protein subunits. The fact that viruses assemble around their own RNA rather than the host mRNa has remained a mystery despite a huge number of theoretical and experimental studies. The presence of the strong attractive interaction between packaging signals and capsid proteins in certain viruses such MS2 and STNV has been identified as the reasons for which capsid proteins selectively package their cognate RNAs over the host mRNA.

In this paper, using stochastic modeling, the authors find that in the case of assembly of STNV, the combined effect of the packaging signal and the specific set of local rules for the RNA connecting capsid proteins significantly increases the assembly efficiency. Based on the experimental data, the authors consider that RNA containing 30 packaging signals connects 60 capsid proteins that make up the capsid. The authors find that only certain local rules are compatible with highly efficient assembly.

I think that the subject is very interesting. The authors are working on a very hot topic. The paper is relatively well-written. The findings of the authors are very interesting; thus, I recommend it for publication. However, I suggest the authors clarify several things before the publication of the paper so a broader audience can benefit from the paper.

1. The authors should explain, at least partially, which set of local rules they plan to use in the introduction of the paper. The introduction appears vague without a statement about the local rules. Maybe they can describe what the rules are about.

2. I understand that the authors argue that the presence of packaging signals significantly increase the efficiency of assembly of STNV. I agree with them. However, the reason that the genome and capsid proteins attract each other is electrostatic interactions. The driving force for the assembly cannot be anything but the electrostatic interaction. The packaging signal can promote the interaction and make it more efficient. I think that the authors should make it clear that the source of attraction between the genome and capsid proteins is electrostatic interaction. Maybe due to the steric interaction, a growing shell remains incomplete. Nevertheless, the reason for the aggregation of capsid proteins on the genome is electrostatics.

3. The N-terminal domains of capsid proteins are positively charged and as such they repel each other. Thus, the electrostatic interaction between the negative charges on RNA and positive charges on N-terminals is absolutely necessary for assembly, see Phys. Rev. E 96, 022401 (2017). Do the authors consider this affinity in their model?

4. The authors state, “in the plant viruses BMV and CCMV [9], the spontaneous encapsulation of ssRNA is driven largely by charge neutralisation interactions …” However, the viruses are almost always overcharged. The number of negative charges on RNA is always higher than the positive charges on capsid proteins, see for example Phys. Rev. E 94, 022408, (2016).

5. RNA assumes different conformations inside the viral shell depending on the interaction between RNA and capsid proteins. Can the authors’ model explain the different conformations of RNA observed in viral capsids, see Phys. Rev. Lett. 119, 188102 (2017).

6. The relation between 3b and 3c is not very clear. I suggest that the authors expand the caption to make it clearer. It also needs to be fixed. There is no part c.

7. I think that the authors need to spend more time to explain their moves. Without knowing the previous papers of the authors, it is hard to follow the set of moves the way it is described in the paper.

8. The authors mention that there are many theoretical works about the non-specific interaction, but they do not refer to almost none of them. See for example Phys. Rev. E 78, 051915 (2008) and the review article, Physics Reports 847, 1-102 (2020) and many references therein.

9. Can the authors explain why only certain local rules are efficient? Is there a physical reason for this?

10. Line 60 refers to Fig. 2d. I think that this is a typo. There is no Fig. 2d.

Reviewer #2: see attachment

Reviewer #3: The article is well written although for non-mathematicians there are some concepts that are hard to understand. Nonetheless, this newer version of Eric and Reydun´s work includes cases in which assembly might not bee mediated by packaging signals. The only improvement that I would like to see, perhaps in a second articles, is what happens when yo take this model and now you also consider a distribution of the number and location of PS. What if an RNA has less PSs but there are spatially located different (closer to each each other) than an RNA that has more PS but they are located far apart from each other. This question is motivated in the case of HIV-1; please see the current version of HIV-1 PS (e.g. Kevin Weeks´s, Alan Rein´s, Wei-Shau Hu´s and Micheal Summer´s work in the past 6 years). There it seems like there are multiple PS located withing a very small portion of the genome.

Anyway, this is a great article.

**Have the authors made all data and (if applicable) computational code underlying the findings in their manuscript fully available?**

Reviewer #1: Yes

Reviewer #2: **No: **The data for the cryo-EM analysis upon which the simulations are based is not deposited in the PDB

Reviewer #3: Yes

PLOS authors have the option to publish the peer review history of their article (what does this mean?). If published, this will include your full peer review and any attached files.

Reviewer #1: No

Reviewer #2: No

Reviewer #3: **Yes: **Mauricio Comas-Garcia

Figure Files:

Data Requirements:

Reproducibility:

References:

---

## [Decision Letter · Decision Letter 1]

29 Jun 2021

Dear Dr. Dykeman,

Thank you very much for submitting your manuscript "The impact of local assembly rules on RNA packaging in a T=1 satellite plant virus" for consideration at PLOS Computational Biology. As with all papers reviewed by the journal, your manuscript was reviewed by members of the editorial board and by several independent reviewers. The reviewers appreciated the attention to an important topic. Based on the reviews, we are likely to accept this manuscript for publication, providing that you modify the manuscript according to the review recommendations.

Sincerely,

Shi-Jie Chen

Associate Editor

PLOS Computational Biology

Ilya Ioshikhes

Deputy Editor

PLOS Computational Biology

[LINK]

Reviewer's Responses to Questions

**Comments to the Authors:**

Reviewer #1: The authors have addressed all the issues and comments that I had raised. To this end, I think that the paper is suitable for publication now.

Reviewer #2: The revised manuscript explains points 1,2,4,5, and 6 much better. This topic is important, and the manuscript provides new insights into virus assembly. However, the authors do not address the most significant concern about basing the simulations and analysis on an unpublished cryoelectron structure showing additional density for the RNA that is not present in other deposited structures for the STNV in the Protein Data Bank. The authors do refer to PDB ID# 4v4m, but this structure does not include density for the RNA. In addition, the authors do not provide more information about the genetic algorithm, scoring criteria, or benchmarking data, as requested in point 3. The authors should consider making the algorithm and its associated datasets available on github or similar open access repository.

**Have the authors made all data and (if applicable) computational code underlying the findings in their manuscript fully available?**

Reviewer #1: Yes

Reviewer #2: **No: **Reference 17 is the only source for the cryoelectron microscopy structure on which the analysis is based. The authors have not described the development of the genetic algorithm or made it publicly accessible.

PLOS authors have the option to publish the peer review history of their article (what does this mean?). If published, this will include your full peer review and any attached files.

Reviewer #1: No

Reviewer #2: No

Figure Files:

Data Requirements:

Reproducibility:

References:

---

## [Decision Letter · Decision Letter 2]

20 Jul 2021

Dear Dr. Dykeman,

Thank you very much for submitting your manuscript "The impact of local assembly rules on RNA packaging in a T=1 satellite plant virus" for consideration at PLOS Computational Biology. As with all papers reviewed by the journal, your manuscript was reviewed by members of the editorial board and by several independent reviewers. The reviewers appreciated the attention to an important topic. Based on the reviews, we are likely to accept this manuscript for publication, providing that you modify the manuscript according to the review recommendations.

Sincerely,

Shi-Jie Chen

Associate Editor

PLOS Computational Biology

Ilya Ioshikhes

Deputy Editor

PLOS Computational Biology

[LINK]

Reviewer's Responses to Questions

**Comments to the Authors:**

Reviewer #2: The concern about the cryoelectron microscopy data refers to the asymmetric reconstruction in reference 17 by J.A. Geraets at the University of York. The 4BCU PDB structure used symmetry averaging to analyze the data. Is the asymmetric reconstruction referred to in reference 17 by Geraets the same data with a new asymmetric analysis of the data in 4BCU or are new data, new maps, or new models generated? Perhaps the model from the asymmetric analysis could be added as an update to the 4BCU submission? Could a brief outline of how the data was re-analyzed be provided? The specific sentences that seem to be the point of miscommunication are repeated below. Perhaps the authors could rephrase the first few sentences in this paragraph to be more clear about exactly what data and models were used to determine local assembly rules.

Line 136-141 R1

“Recently, a cryo-EM asymmetric reconstruction of the STNV capsid and its RNA density was determined which revealed potential ways in which the RNA strand could connect between neighbouring PSs in contact with the capsid [17]. Based on this asymmetric cryo-EM reconstruction, we determine the local assembly rules i.e., the possible ways that a CP:PS complex that is 5’ (or 3’) to the growing capsid could associate to the partially formed capsid.”

17. Geraets JA. Self-assembling nanoscale systems. University of York; 2015.

Lines 136-140 R2

“Recently, an analysis of the STNV capsid and the RNA contacts seen in Ford et al. [14] revealed potential ways in which the RNA strand could connect between neighbouring PSs in contact with the capsid [17]. Based on this analysis, we determine the local assembly rules i.e., the possible ways that a CP:PS complex that is 5’ (or 3’) to the growing capsid could associate to the partially formed capsid.”

14. Ford RJ, Barker AM, Bakker SE, Coutts RH, Ranson NA, Phillips SE, et al. Sequence-specific, RNA–protein interactions overcome electrostatic barriers preventing assembly of satellite tobacco necrosis virus coat protein. Journal of Molecular Biology. 2013;425(6):1050–1064.

17. Geraets JA. Self-assembling nanoscale systems. University of York; 2015.

The authors’ efforts to make more information about the genetic algorithm available is appreciated. The website link to the assembly code could be included in the manuscript. The link provided in the authors’ response (https://www-users.york.ac.uk/~ecd502/software.html) goes to the Dykeman research web site and has links to 3 software programs, Saguaro Biomolecular Simulation Package, KFOLD, and Ribosome Translation Kinetics. The authors might make it more explicitly clear how to access the assembly code. Alternatively, a site such as Figshare or github provides a public space to deposit in-house code and scripts and make them easily available to others.

**Have the authors made all data and (if applicable) computational code underlying the findings in their manuscript fully available?**

Reviewer #2: **No: **Please make the asymmetric reconstruction and scripts publicly accessible on a site such as github or figshare.

PLOS authors have the option to publish the peer review history of their article (what does this mean?). If published, this will include your full peer review and any attached files.

Reviewer #2: No

Figure Files:

Data Requirements:

Reproducibility:

References:

---

## [Editor Report · Decision Letter 3]

26 Jul 2021

Dear Dr. Dykeman,

We are pleased to inform you that your manuscript 'The impact of local assembly rules on RNA packaging in a T=1 satellite plant virus' has been provisionally accepted for publication in PLOS Computational Biology.

Best regards,

Shi-Jie Chen

Associate Editor

PLOS Computational Biology

Ilya Ioshikhes

Deputy Editor

PLOS Computational Biology

---

## [Editor Report · Acceptance letter]

9 Aug 2021

PCOMPBIOL-D-21-00604R3 

The impact of local assembly rules on RNA packaging in a T=1 satellite plant virus

Dear Dr Dykeman,

I am pleased to inform you that your manuscript has been formally accepted for publication in PLOS Computational Biology. Your manuscript is now with our production department and you will be notified of the publication date in due course.

With kind regards,

Andrea Szabo
